# PIF transcription factors link a neighbor threat cue to accelerated reproduction in *Arabidopsis*

Vinicius Costa Galvão [1], Anne-Sophie Fiorucci [1], Martine Trevisan [1], José Manuel Franco-Zorilla[2], Anupama Goyal[1,4], Emanuel Schmid-Siegert [3], Roberto Solano [2] & Christian Fankhauser [1]

Changes in light quality indicative of competition for this essential resource influence plant growth and developmental transitions; however, little is known about neighbor proximity-induced acceleration of reproduction. Phytochrome B (phyB) senses light cues from plant competitors, ultimately leading to the expression of the floral inducers *FLOWERING LOCUS T* (*FT*) and *TWIN SISTER of FT* (*TSF*). Here we show that PHYTOCHROME INTERACTING FACTORs 4, 5 and 7 (PIF4, PIF5 and PIF7) mediate neighbor proximity-induced flowering, with PIF7 playing a prominent role. These transcriptional regulators act directly downstream of phyB to promote expression of *FT* and *TSF*. Neighbor proximity enhances PIF accumulation towards the end of the day, coinciding with enhanced floral inducer expression. We present evidence supporting direct PIF-regulated *TSF* expression. The relevance of our findings is illustrated by the prior identification of *FT*, *TSF* and *PIF4* as loci underlying flowering time regulation in natural conditions.

[1] Center for Integrative Genomics, Faculty of Biology and Medicine, University of Lausanne, 1015, Lausanne, Switzerland. [2] Genomics Unit and Plant Molecular Biology Department, Centro Nacional de Biotecnologia (CSIC), Campus de Cantoblanco, Darwin 3, 28049 Madrid, Spain. [3] SIB Swiss Institute for Bioinformatics, University of Lausanne, 1015, Lausanne, Switzerland. [4]Present address: Syngene International Ltd, Bangalore 560 099, India. Correspondence and requests for materials should be addressed to C.F. (email: christian.fankhauser@unil.ch)

Plants depend on sunlight to fuel photosynthesis. Therefore, growing with potentially reduced light availability, as encountered in dense plant communities, constitutes a threat for plant growth and development. Plants perceive potential competitors because of the reflected far-red (FR) light from neighbors, resulting in reduced red (R)/FR ratio (low R/FR), which leads to the conversion of active phytochrome (Pfr) photoreceptors to their inactive Pr form[1].

In shade-intolerant plants, such as *Arabidopsis thaliana*, neighbor detection triggers organ elongation to outgrow competitors and precocious flowering[1]. Among the five phytochromes (phys) present in *Arabidopsis*, phytochrome B (phyB) is the major regulator of shade avoidance[1]. In normal light conditions, photoactive phyB interacts with a class of basic helix–loop–helix (bHLH) transcription factors, the PHYTOCHROME-INTERACTING FACTORS (PIFs), leading to their phosphorylation and degradation via 26S proteasome or inactivation through less understood mechanisms[2,3]. In contrast, in low R/FR, PIF phosphorylation and degradation is prevented due to the reduction in photoactive phyB enabling PIF binding to G-boxes and PBE-boxes in the promoters of many shade-regulated genes[3–10]. We have started to understand how these transcriptional events lead to shade-induced growth responses[1]; however, we know very little about developmental transitions triggered by neighbor proximity.

Accelerated flowering is one of the most dramatic responses in the presence of neighboring plants, particularly for annual plants where it represents a unique and irreversible event. In *Arabidopsis*, accelerated flowering in low R/FR results from the transcriptional induction of *FLOWERING LOCUS T* (*FT*) and its close homolog *TWIN SISTER OF FT* (*TSF*) in the phloem companion cells of the leaf vasculature[11–14]. FT acts as a long-distance florigen signal being transported to the shoot apical meristem (SAM) where it forms a flowering activation complex (FAC) after interacting with the basic leucine zipper (bZIP) transcription factor FD and the 14-3-3 proteins[15–17]. At the SAM the FAC induces flowering by activating the expression of flowering time genes, such as the MADS-box *SUPPRESSOR OF OVER-EXPRESSION OF CO 1* (*SOC1*) and *APETALA1* (*AP1*)[16,17].

Despite their importance for shade-induced flowering, the mechanism linking low R/FR ratio perception by phys to enhanced *FT* and *TSF* expression in the vasculature remains poorly understood. In *Arabidopsis*, low R/FR promotes floral transition in a photoperiod-dependent manner[18] in agreement with the attenuated low R/FR response of the photoperiodic mutant *constans* (*co*)[11,18]. PHYTOCHROME AND FLOWERING TIME 1 (PFT1) was initially proposed to control flowering in response to simulated shade[13], but was later shown to respond normally to continuous low R/FR[18,19]. Here we show that a subset of PIF transcription factors function genetically downstream of phys to regulate flowering time through *FT* and *TSF* expression in response to low R/FR.

## Results

### PIFs control low R/FR-induced flowering downstream of phyB.
PIFs play major roles in neighbor detection growth responses downstream of phyB[4,20,21]. Enhanced *PIF* expression induces precocious flowering through *FT* and *TSF* in the phloem[22–24]. Moreover, plants with impaired HFR1 function, a repressor of PIF activity[25], display increased *FT* expression in response to low R/FR[26]. Therefore, we hypothesized that PIFs might control flowering time in response to low R/FR. To test this, we scored the flowering transition of *PIF* loss-of-function mutants under simulated neighbor proximity conditions (Supplementary Fig. 1a, b; hereafter referred to as low R/FR, while high R/FR refers to

standard condition) and found that PIF7 plays a prominent function to accelerate floral transition under low R/FR (Fig. 1a–c; Supplementary Fig. 2a, b). In addition, mutations in *PIF4* and *PIF5* further enhanced the *pif7* late flowering phenotype, indicating that these genes also contribute to the response (Fig. 1a–c; Supplementary Figs. 2a, b and 3a–c). Importantly, while *pif4 pif5 pif7* flowered slightly later than the wild type in high R/FR, the *pif4 pif5 pif7* phenotype was much enhanced in low R/FR (significant interaction between genotype and condition both in terms of days to flowering ($p = 0.0007$) and leaf number ($p = 1.52e − 05$)) showing the requirement for those three PIF particularly for accelerated flowering in low R/FR (Fig. 1c; Supplementary Fig. 2b). Moreover, while *pif7*, *pif3 pif4 pif5*, and *pif4 pif5 pif7* mutants flowered slightly later than the wild type in high R/FR, in low R/FR late flowering was specifically observed in *pif7* and *pif4 pif5 pif7* mutants (Fig. 1a–c; Supplementary Fig. 3a–c). Next, we checked whether PIFs mediate precocious flowering of the constitutive shade-avoidance mutant *phyB*. Consistent with our data in low R/FR, mutations in *PIF4*, *PIF5*, and *PIF7* were required to fully suppress early flowering in *phyB* in high R/FR in inductive (long days (LDs)) and non-inductive (short days (SDs)) photoperiods (Fig. 1d, e; Supplementary Fig. 4a–d). Finally, because low R/FR further accelerates flowering in *phyB* mutant due to the activity of other phys[27,28] (Supplementary Fig. 5a–c), we scored *phyB pif4 pif5 pif7* flowering in low R/FR. In this condition, *phyB pif4 pif5 pif7* flowers later than *phyB*, at the same time as *pif4 pif5 pif7* (Supplementary Fig. 5a–c), further supporting the role of PIF4, PIF5, and PIF7 in the phyB pathway. We conclude that PIF4, PIF5, and PIF7 act genetically downstream of phyB, and possibly other phys, to control low R/FR-induced flowering.

### PIFs control *FT* and *TSF* expression in response to low R/FR.
Because flowering in low R/FR depends on both the growth condition and the genetic background[11,18,27], we tested the flowering response of *ft*, *tsf*, *ft tsf*, and *co*. In our conditions *ft* and *tsf* single mutants responded strongly to low R/FR[11,29] (Supplementary Fig. 6a–d), whereas *ft tsf* double mutants presented a reduced low R/FR response, similar to *co* (Supplementary Fig. 6a–d), suggesting that FT and TSF together are needed to promote flowering in low R/FR (significant interaction between genotype (*ft* vs. *ft tsf*) and condition both in terms of days to flowering ($p = 8.7e − 05$) and leaf number ($p < 2e − 16$)) (Supplementary Fig. 6d).

To compare the role of FT and TSF in the constitutive shade-avoidance mutant *phyB* with plants growing in low R/FR, we scored the flowering phenotype of *phyB ft*, *phyB tsf*, and *phyB ft tsf*. While *phyB tsf* double mutants flowered as *phyB*, both *ft* and *ft tsf* abolished *phyB* early flowering in standard conditions (Supplementary Fig. 6e). Together, these findings indicate that the *phyB* mutant does not phenocopy plants growing in low R/FR in terms of flowering, likely because other phys also act in low R/FR. This is consistent with acceleration of flowering by low R/FR in the *phyB* mutant[27] (Supplementary Fig. 5a–c) and the extreme early flowering of *ft* when combined with higher-order phytochrome mutants[30,31].

Next, we determined whether PIFs contribute to *FT* and *TSF* transcriptional regulation in low R/FR. Transcriptome data[32] showed that *FT* messenger RNA (mRNA) levels increased in cotyledons within 90 min after transfer to low R/FR, while such a rapid induction was not observed for *TSF* (Supplementary Fig. 7a, b). In contrast, this early *FT* up-regulation was absent in the *pif4 pif5 pif7* mutant (Supplementary Fig. 7c). We therefore monitored *FT* and *TSF* expression in the wild type and *pif4 pif5 pif7* for several days after transfer from high to low R/FR at

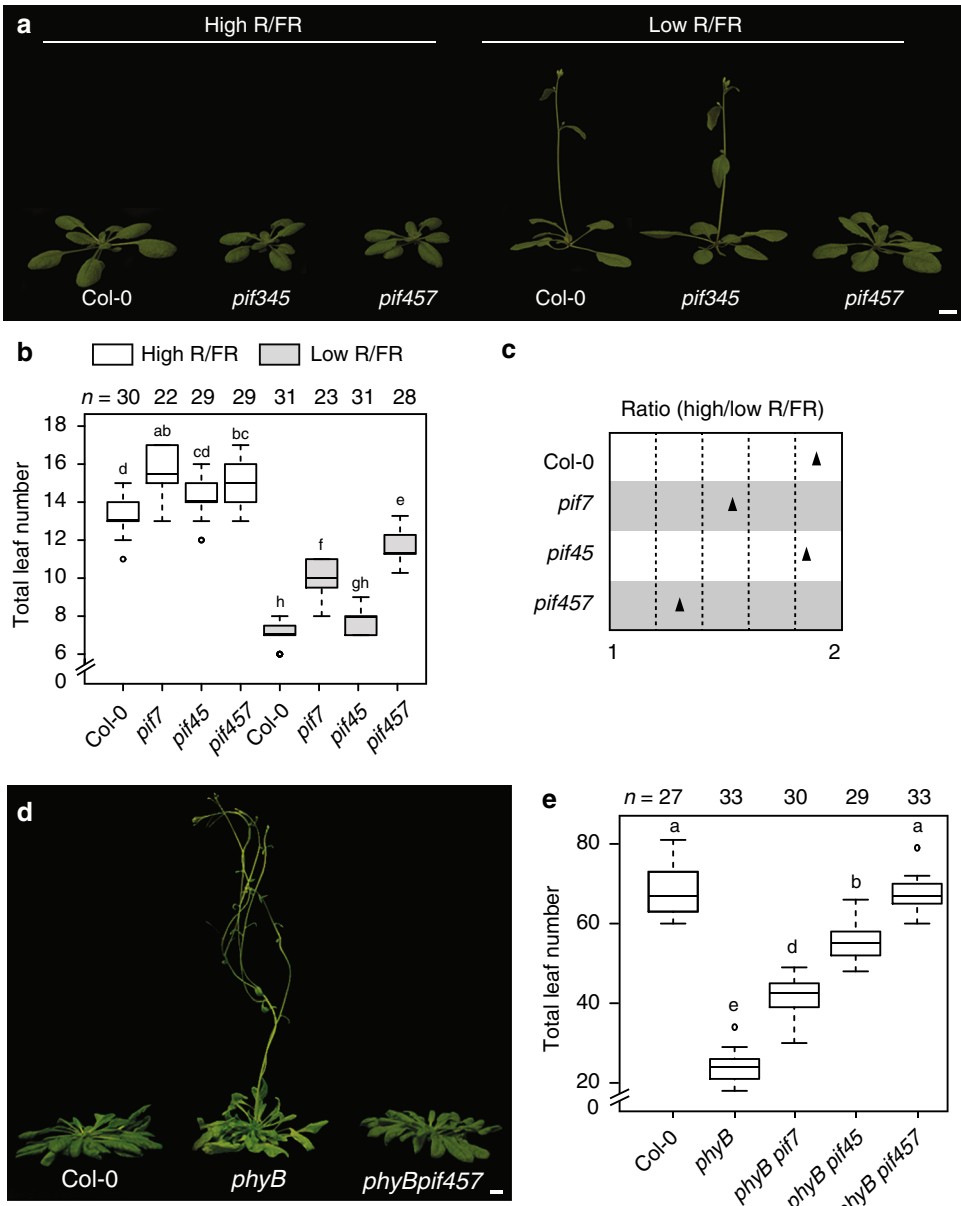

**Fig. 1** PHYTOCHROME-INTERACTING FACTORS (PIFs) mediate flowering in the low red/far-red (R/FR) downstream of phytochrome B (phyB). Plants were grown for 5 days in high R/FR for complete de-etiolation and either kept in high R/FR or shifted to low R/FR on day 6 until the onset of flowering. **a** Twenty-two-day-old Col-0, *pif3 pif4 pif5* and *pif4 pif5 pif7* grown under long days (LDs) at 22 °C in high and low R/FR with flowering phenotype represented as total leaf number **b**, **c** Leaf number ratio (high vs. low R/FR) of plants phenotyped in **a**. **d** Representative image of 53-day-old Col-0, *phyB* and *phyB pif4 pif5 pif7* under short days (SDs) at 22 °C in high R/FR and flowering phenotype represented as total leaf number **e** Boxplots were created using the online BoxPlotR[80]; center line, median; box limits, upper and lower quartiles; whiskers, 1.5× interquartile range (IQR); dots, outliers. *n* represents the number of plants phenotyped. Letters represent the significance groups at *p* value <0.01 using one-way analysis of variance (ANOVA), followed by Tukey's honestly significant difference (HSD) test. Scale bar correspond to 1 cm

ZT16, as *FT* and *TSF* expression peak at dusk[18,33]. *FT* and *TSF* expression were similar in *pif4 pif5 pif7* and wild-type plants in high R/FR. In contrast, *FT* and *TSF* up-regulation by low R/FR were strongly impaired in *pif4 pif5 pif7* (Fig. 2a, b). Consistent with the importance of PIF-dependent *TSF* up-regulation, *ft pif4 pif5 pif7* quadruple mutants flowered later compared to *ft* and similar to *ft tsf* under low R/FR (Supplementary Fig. 8a–d). Moreover, in low R/FR *TSF* expression was reduced in *ft pif4 pif5 pif7* compared to the *ft* single mutant (Supplementary Fig. 8e). Low R/FR-mediated induction of *FT* and *TSF* were reduced in *pif7* single mutant and further reduced in *pif4 pif5 pif7* (Fig. 2c, d), correlating with the flowering time defects of those mutants

(Fig. 1b, c; Supplementary Fig. 2). Similarly, the constitutively higher expression of *FT* and *TSF* in *phyB* was gradually reduced in higher-order *pif* mutants (Supplementary Fig. 9a, b). Collectively, these results show that PIF4, PIF5, and PIF7 contribute to low R/FR-mediated floral transition, as well as *FT* and *TSF* expression.

In contrast to *FT* and *TSF*, *CO* mRNA expression was only marginally increased by low R/FR light treatments in both wild type and the *pif4 pif5 pif7* mutant (Supplementary Fig. 10). In order to determine whether PIF4, PIF5, and PIF7 control low R/FR-induced flowering exclusively through the CO-FT/TSF pathway, we compared flowering of *co* and *co pif4 pif5 pif7* mutants.

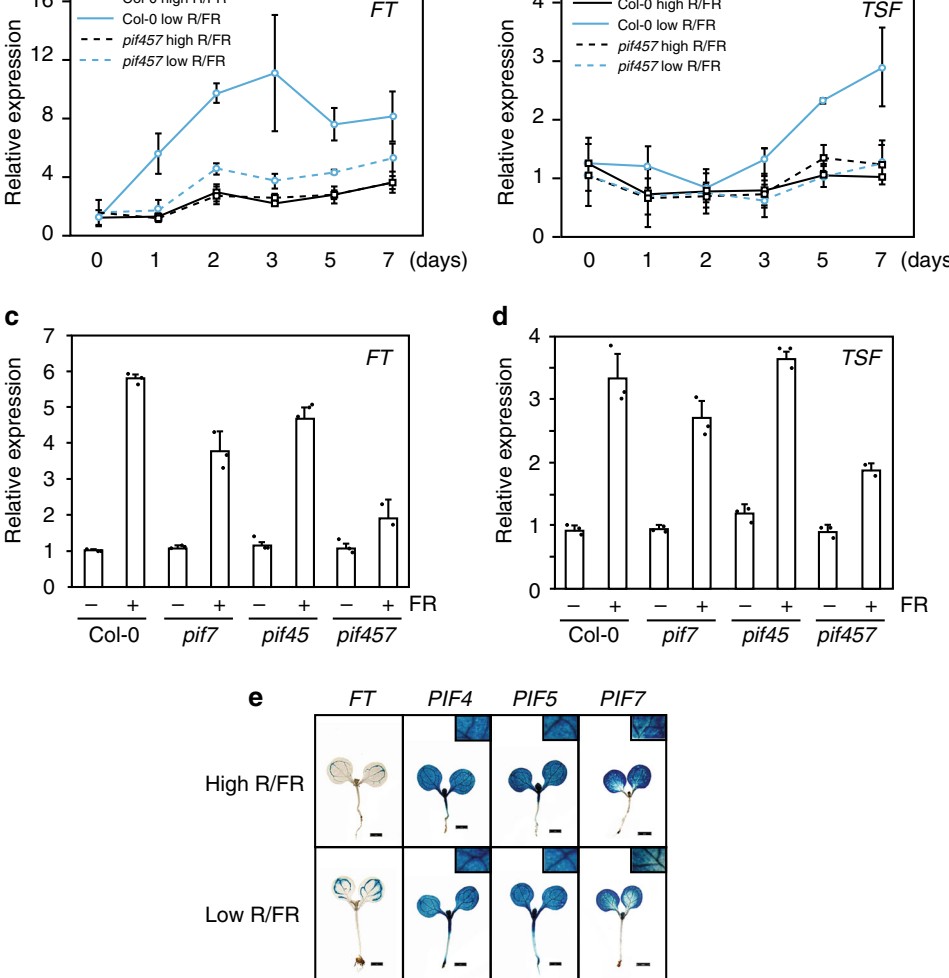

**Fig. 2** *PHYTOCHROME-INTERACTING FACTORS* (*PIF*) proteins mediate *FLOWERING LOCUS* (*FT*) and *TWIN SISTER of FT* (*TFS*) expression in the low red/far-red (R/FR). **a**, **b** *FT* and *TSF* messenger RNA (mRNA) level using quantitative real-time PCR (RT-qPCR) after shift from high to low R/FR in Col-0 and *pif4 pif5 pif7*. Plants were de-etiolated for 5 days under long days (LDs) at 22 °C in high R/FR and samples were harvested at zeitgeber (ZT) 15–6 before 0 and 1, 2, 3, 5, and 7 days after transfer to low R/FR. Error bars represent 2× SEM. **c**, **d** *FT* and *TSF* mRNA levels of plants growing under LDs at 22 °C in high and low R/FR. Plants were grown for 5 days in high R/FR for complete de-etiolation. On day 6, plants were either shifted to low R/FR (FR+) or kept in high R/FR (FR−). Samples were harvested 10 days after sowing at ZT 15–16. Error bars represent standard deviation of three biological and three technical replicates, individual data points are indicated with black dots. **e** β-Glucuronidase (GUS) staining of *pPIF4::GUS*, *pPIF5::GUS*, and *pPIF7::GUS* transgenic seedlings 7 days after sowing. Plants were grown on soil for 5 days under LDs at 22 °C in high R/FR and either kept in the same condition of shifted to low R/FR on day 6. Samples were harvested after GUS staining at ZT 15–16. Scale bar corresponds to 1 mm

Interestingly, the quadruple mutant flowered later than *co* both in high and low R/FR, indicating that the PIFs also control flowering through a CO-independent pathway (Supplementary Fig. 11a–c). Moreover, similar to *ft pif4 pif5 pif7* and *ft tsf*, the quadruple *co pif4 pif5 pif7* responded to low R/FR, suggesting that independent flowering pathways mediate low R/FR-induced flowering (Supplementary Figs. 8a–c and 11a–c). Collectively, our results identify PIFs as important mediators of FT- and TSF-induced early flowering in response to low R/FR and suggest that PIFs also control flowering through a CO-FT/TSF-independent pathway.

**PIF protein accumulation correlates with *FT-TSF* expression.** To better understand how PIFs control *FT* and *TSF* expression, we investigated their temporal and spatial expression pattern. Consistent with the vascular expression of *FT* and *TSF* during floral transition[11,12,34] (Fig. 2e), promoter-GUS (β-glucuronidase) fusions showed broad *PIF4*, *PIF5*, and *PIF7* expression, including

the leaf vasculature, in seedlings (Fig. 2e). This is consistent with tissue-specific expression analysis of *PIF4*, *PIF5*, and *PIF7*[35,36], indicating that *PIF4*, *PIF5*, *PIF7*, *FT*, and *TSF* are expressed in the vasculature. *FT* mRNA expression in the wild type displayed two strong peaks in response to low R/FR, the first early in the light period and the highest peak around dusk, and a third smaller peak during the night (Fig. 3a)[18]. In contrast, there was no induction of *FT* expression by low R/FR in *pif4 pif5 pif7* (Fig. 3a). The *TSF* diel expression pattern and its regulation by low R/FR and the PIFs were very similar to *FT* (Fig. 3a, b). In contrast to *FT* and *TSF*, *PIF4*, *PIF5*, and *PIF7* expression showed one strong peak in the morning as previously observed for *PIF4* and *PIF5*[37] (Fig. 3c–e). Low R/FR ratio led to a slower decline from peak levels that were particularly obvious for *PIF4* and *PIF5*, but not significant for the lower amplitude cycling *PIF7* gene (Fig. 3c–e). This expression pattern is consistent with a previous study, which showed that low R/FR slows down the circadian clock[38]. Given that phyB inactivation under low R/FR stabilizes PIF4 and PIF5

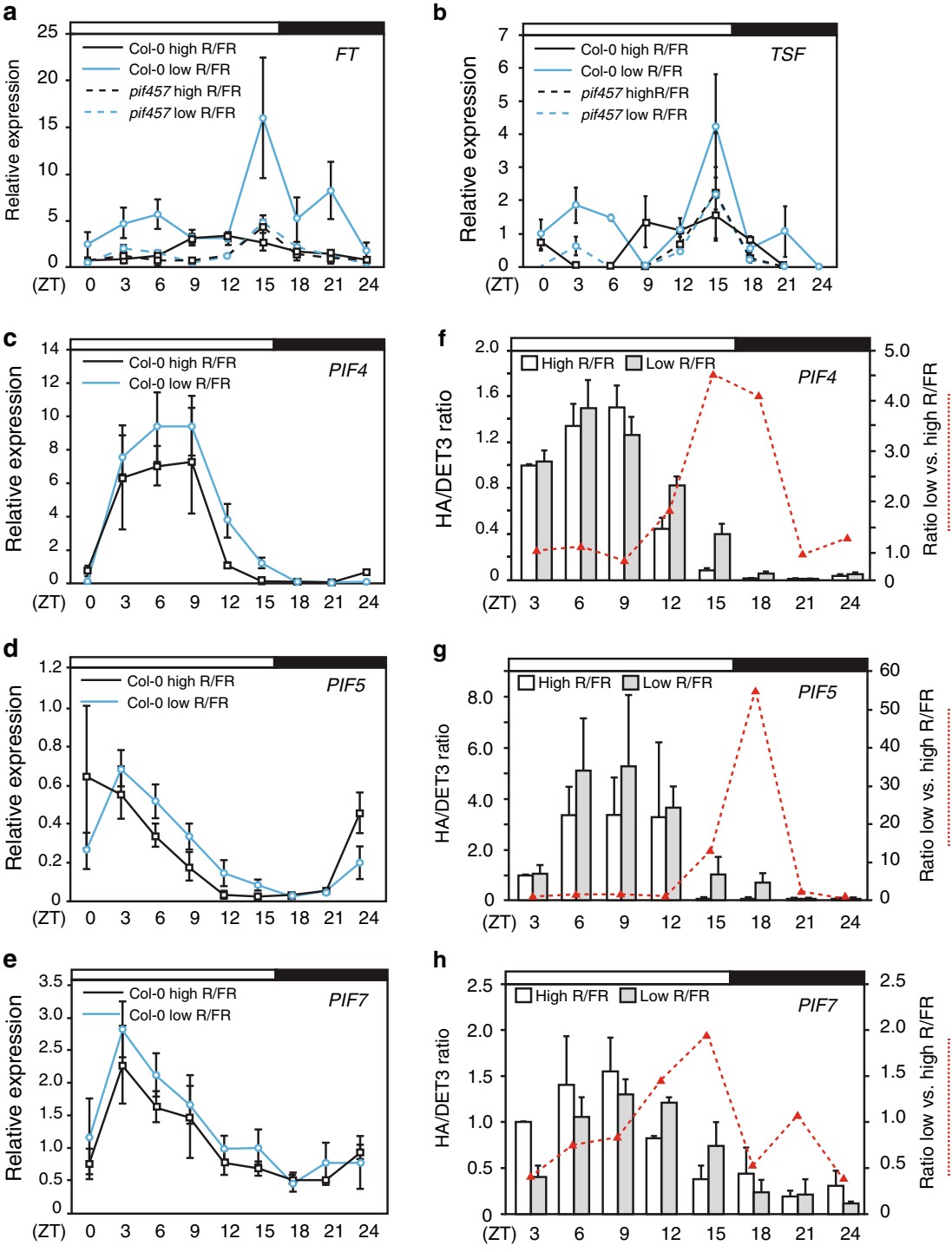

**Fig. 3** PHYTOCHROME-INTERACTING FACTORS (PIFs) and *FLOWERING LOCUS/TWIN SISTER of FT* (*FT/TSF*) expression pattern in high and low red/far-red (R/FR) ratio. Diel time course of *FT* (**a**), *TSF* (**b**), *PIF4* (**c**), *PIF5* (**d**), and *PIF7* (**e**) over 24 h harvested every 3 h. Expression levels were determined using quantitative real-time PCR (qRT-PCR) using three biological replicates and three technical replicates of 10–11-day-old Col-0 and *pif4pif5pif7* mutant. Plants were grown under high R/FR for 5 days and either kept in the same condition of shifted to low R/FR on day 6. White and dark bars on top of each chart correspond to light and dark phases, respectively. Error bars correspond to 2× standard error. Protein accumulation of *pif4-101/pPIF4::PIF4-3HA* (**f**), *pif5-3/pPIF5::PIF5-3HA* (**g**), and *pif7-2/pPIF7::PIF7-3HA* (**h**) in 10–11-day-old plants harvested every 3 h. White (high R/FR) and gray (low R/FR) bars correspond to the average protein levels of three biological replicates and at least two technical replicates relative to DET3. Red dashed lines represent the PIF protein level ratio of low/high R/FR. Error bars represent standard deviation and white and black bars on top of each chart represent the light and dark phases, respectively

proteins[4], we decided to also investigate PIF4, PIF5, and PIF7 protein accumulation. Using genomic HA-tagged lines driven by their own promoters (Supplementary Fig. 12)[39], we observed diel protein oscillation of HA-tagged PIF4, PIF5, and PIF7 matching mRNA levels (Fig. 3f–h; Supplementary Fig. 13). Moreover, as reported previously[20], upon transfer to low R/FR we also observed a change in the ratio of slower and faster migrating PIF7 isoforms likely corresponding to different phosphorylation forms (Supplementary Fig. 14). In addition, we noted a transient increase in total PIF7 levels in low R/FR (Supplementary Fig. 14). Interestingly, PIF4 and PIF5 proteins accumulated to higher levels in low R/FR specifically toward the end of the day, correlating with FT and TSF expression (Fig. 3a, b, f, g; Supplementary Fig. 13). There was also a tendency for enhanced end-of-day accumulation of PIF7 in low R/FR, but it was less pronounced compared to PIF4 and PIF5, as also observed for the RNA expression patterns (Fig. 3f–h; Supplementary Fig. 13). However, PIF7 nuclear import is induced by low R/FR[3], indicating that a different mode of low R/FR regulation may be involved for PIF7. Multiple levels of PIF regulation by low R/FR is further suggested by the PIF-dependent expression of FT and TSF in low R/FR early in the day, a time at which no significant difference in PIF4, PIF5, or PIF7 accumulation was observed (Fig. 3a, b, f–h).

**PIF proteins directly control TSF expression.** PIF4 and PIF5 preferentially bind to G-boxes (CACGTG) and PBE-boxes (CATGTG)[8,9]. Because PIF7 plays a central role in low R/FR-induced flowering and little is known about its DNA-binding preference, we tested its DNA-binding specificity using protein-binding microarrays. In agreement with recent DNA affinity purification sequencing (DAP-seq) data[40], and similar to other PIFs[8,9], we found that PIF7 binds with high affinity to G-boxes (Fig. 4a). Moreover, as observed for other PIFs, among E-boxes it showed the highest affinity for the PBE-box (Fig. 4a).

Interestingly, the analysis of previously published chromatin immunoprecipitation-sequencing (ChIP-seq) data[10] revealed a high-confidence PIF4 binding peak containing 1 G-box and 1 PBE-box overlapping with a highly conserved region located 1.5 kb downstream of FT stop codon (Supplementary Fig. 15)[41]. Because TSF was shown to integrate environmental signals to influence flowering time in natural conditions[42] and TSF is clearly important for low R/FR-induced flowering in our conditions (Supplementary Fig. 6a–d), we focused our analysis on PIF regulation of TSF expression. We identified no G-boxes, but identified two PBE-boxes in the TSF promoter located 990 and 437 bases upstream of the ATG (Fig. 4b). The analysis of ChIP-seq data[10] revealed a high-confidence PIF4 peak overlapping with the first PBE-box (−437) (Fig. 4b). To test whether these PBE-boxes are biologically relevant for PIF-mediated TSF expression, we fused its promoter (wild-type and PBE mutants) with luciferase and performed transient transactivation assays in Nicotiana benthamiana. Consistent with PIFs directly regulating TSF expression, PIF4 and PIF7 led to TSF expression (Fig. 4c, d). Importantly, TSF expression was almost completely abolished by a single-nucleotide mutation in the PBE-box present in the PIF4 ChIP peak (−437)[10] or by mutations in both PBE-boxes and one CAAGTG sequence (Fig. 4c, d). Taken together, our data suggest that PIF4 and PIF7 directly bind to PBE-boxes at TSF promoter to induce its expression.

## Discussion

Neighbor detection leads to precocious flowering that is controlled via the phyB photoreceptor and FT and TSF genes in Arabidopsis[11,13,14,27,29]. In contrast to well-characterized flowering time pathways, such as photoperiod, vernalization, and

gibberellic acid, how sensing neighbor proximity cues are linked to expression of "florigen" genes remains poorly understood. Our experiments identify PIF4, PIF5, and PIF7 as transcription factors acting downstream of phyB to induce flowering in response to neighbor proximity through the floral inducers FT and TSF (Figs. 1, 2; Supplementary Figs. 2–5, 8, and 9). On the other hand, both our data and a recent study analyzing flowering and FT expression profiles in more natural conditions reveal a limited role for the PIFs in high R/FR[43] (Figs. 1, 2; Supplementary Figs. 2–4). Therefore, our data reveal an important role for PIFs in regulating flowering particularly in response to low R/FR and identify PIF7 as a novel positive regulator of flowering transition.

The perception and response to low R/FR and elevated temperature share sensing and signaling components, including phyB and PIF4[44–46]. However, while the thermal induction of flowering almost completely depends on FT and is mostly independent of CO[47], low R/FR induction of flowering requires CO (and LDs) and depends both on FT and TSF[11,18,29] (Supplementary Fig. 6). Under non-inductive warm SDs, a key flowering-inducing function was initially attributed to PIF4 in directly mediating FT expression[22]. However, both the underlying mechanism and the importance of PIF4 in thermal-induced flowering via FT regulation have been questioned[23,24,48,49]. Here, we show that PIF7 plays the most prominent role in low R/FR-induced flowering with additional roles played by PIF4 and PIF5 (Fig. 1; Supplementary Figs. 2 and 3). The flowering phenotype and FT/TSF expression in pif mutants are both consistent with a role for multiple PIFs in low R/FR-induced flowering through the FT/TSF pathway (Figs. 1, 2; Supplementary Figs. 2–5, 8, and 9). Finally, we present evidence for direct regulation of TSF expression by PIF7 and PIF4, providing a mechanistic basis for low R/FR-induced flowering (Fig. 4).

Our genetic data also indicate that additional mechanisms contribute to the regulation of flowering in low R/FR independently of FT-TSF and PIF proteins. Indeed, similar to co and ft tsf (Supplementary Fig. 6)[11,18,29], co pif4 pif5 pif7 and ft pif4 pif5 pif7 are still responsive to low R/FR (Supplementary Figs. 8 and 11). CO and FT are major flowering time genes expressed in companion cells of leaves and subsequently FT (and possibly TSF) is transported to the SAM to induce flowering[17,34]. Although it could be speculated that low R/FR directly regulate phys and flowering at the SAM, this seems unlikely because it is normally shaded by emerging leaves. Alternatively, other mobile flowering signals, such as gibberellic acid[50,51], could be produced in leaves downstream of phys[52,53] and transported to the SAM to induce flowering[53–57]. Indeed, GA metabolism genes are induced by FR light[52] and the GA-responsive flowering time gene LEAFY (LFY) is up-regulated in the phyB mutant background[53]. Alternatively, low R/FR could directly regulate the miR156-SPL aging pathway[58] independently of GA. PIF proteins directly regulate the expression of MIR156 genes and, consequently, several SPL transcription factors in response to end-of-day FR treatments[59]. However, it should be noted that while miR156-SPL age pathway function mostly at the SAM, the tissue-specific expression of PIFs in this tissue have a modest effect on flowering in Arabidopsis[23]. Although we currently do not understand all the pathways induced by low R/FR to accelerate flowering, we identify the phyB-PIF-FT/TSF regulon as one important mechanism (Fig. 4e).

The phyB mutant is often used as a genetic mimic of low R/FR-grown plants. However, when it comes to the regulation of flowering, we and others observe interesting similarities but also differences between both situations. Early flowering in low R/FR and in phyB requires the activity of several PIFs (Fig. 1; Supplementary Figs. 4 and 5). In contrast, while FT completely accounts for early flowering in phyB (Supplementary Fig. 6e) both in LD or SD photoperiods, low R/FR triggered flowering requires both FT

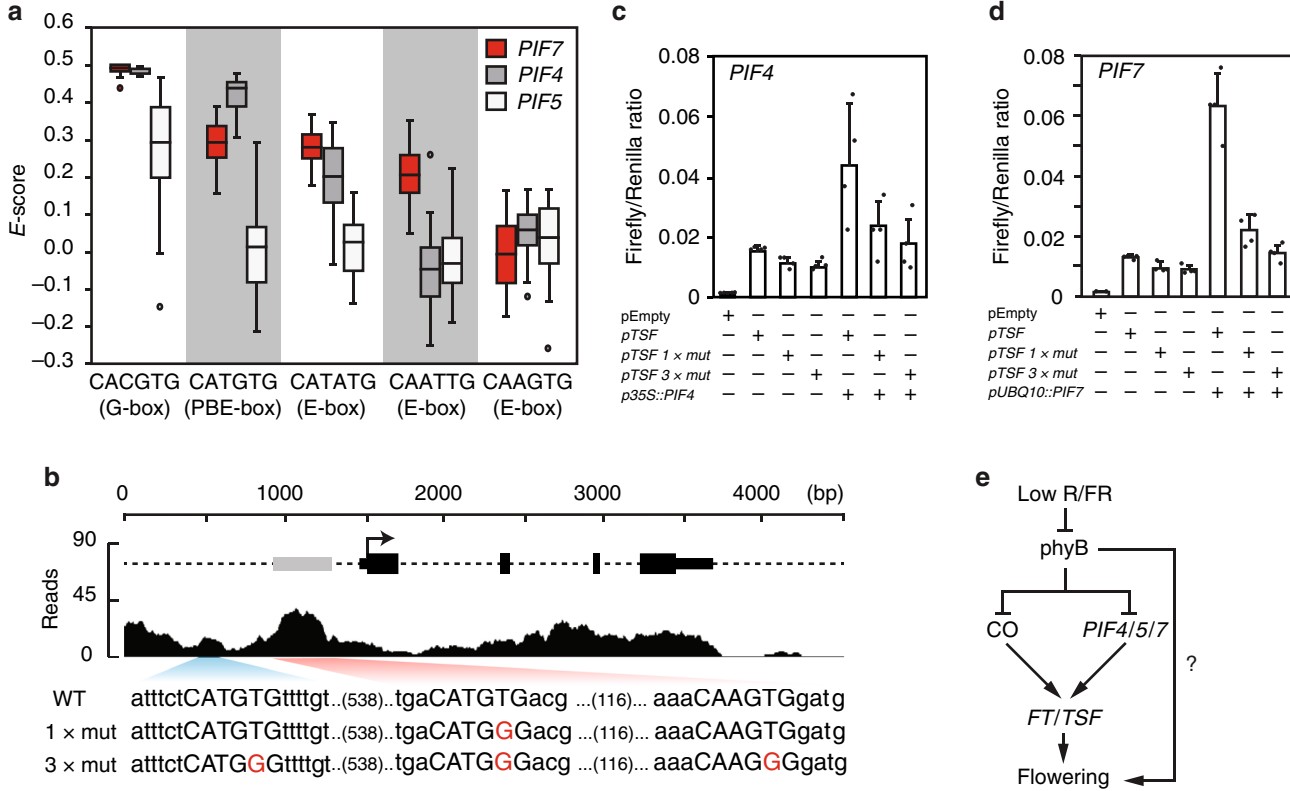

**Fig. 4** PHYTOCHROME-INTERACTING FACTORS (PIF) proteins directly regulate *TWIN SISTER of FT* (*TSF*) expression. **a** PIF7 preferentially bind to G-boxes (CACGTG) and PBE-boxes (CATGTG) in protein-binding microarray using PIF7_bHLH-MBP (Supplementary Table 2). Data corresponding to PIF4 and PIF5 were previously described[8] and shown here for comparison. Box plot represents the distribution of enrichment scores (*E*-scores) for G- and PBE-boxes, as well as for other E-boxes indicated. *E*-score is a rank-based, non-parametric measure of binding affinity that ranges between −0.5 and +0.5 and is calculated from the median intensity of all the oligonucleotide probes that contain a given 8-mer motif[78]. Boxes represent quartiles 25–75%, the black line within represents the median of the distribution (quartile 50%) and dots denote outliers of the distribution. **b** Representation of PIF4 chromatin immunoprecipitation-sequencing (ChIP-seq) reads mapped to the *TSF locus*[10]. Gray box represents high-confidence PIF4 binding peak at *TSF* promoter and nucleotide sequence represents the wild-type (WT) and mutant version containing 1 (1× mut) or 3 mutations (3× mut) used for transient dual-luciferase assays in *Nicotiana benthamiana*. **c**, **d** Luciferase ratio corresponds to the average of *pTSF::fireflyLUC* and *p35S::renillaLUC* ratio of four independent infiltrations and error bars correspond to standard deviation; individual data points are indicated with black dots. **e** Model of low R/FR-regulated flowering time

and TSF and long photoperiods[11,18,29,31] (Supplementary Fig. 6c). The requirement for inductive photoperiods to trigger low R/FR-induced flowering is also consistent with the reduced low R/FR response of *co* and *gi* mutants[18]. An interesting possibility is that the recently reported interaction of CO with PIF4[24], or with other PIFs, could contribute to low R/FR-induced flowering in inductive photoperiods. Indeed, low R/FR leads to increased levels of CO[12], PIF4, and PIF5[4,11] (Fig. 3; Supplementary Figs. 13 and 14). Moreover, the timing of low R/FR-induced PIF levels is noteworthy as the effect is biggest at the end of the day when CO protein accumulates and coincides with enhanced *FT* and *TSF* expression (Fig. 3). Increased levels of PIF4 and PIF5 proteins at the end of the day could be due to both transcriptional and post-transcriptional effects. Indeed, it was previously shown that low R/FR slows down the circadian clock, which may explain the slower decline of *PIF4* and *PIF5* expression towards the end of the day (Fig. 3c, d)[38]. Therefore, both the higher *PIF* transcript levels late in the day and enhanced stability of PIF4 and PIF5 protein in low R/FR[4] may contribute to higher PIF levels coinciding with the late *FT* and *TSF* expression peaks in low R/FR (Fig. 3). Moreover, we note that low R/FR also leads to higher *FT* and *TSF* expression shortly after dawn, a phenomenon that was recently also observed in natural growth conditions (Fig. 3a, b)[43]. Given that at this time of the day we did not observe significant changes in PIF abundance in low vs. high R/FR, we propose that other mechanisms must be operating in shade to control PIF activity. These may include a control of subcellular localization that was reported for PIF7[3] but also direct inhibition of PIF activity by light-activated phyB, which is more abundant in high R/FR[46,60–62]. In summary, the coincidence of high PIF and CO activity towards the end of the day could explain accelerated flowering triggered by neighbor proximity in inductive photoperiods.

In conclusion, multiple mechanisms may contribute to the regulation of PIF protein abundance and activity in low R/FR, but more research is required to fully understand this regulation. However, our data are consistent with the coordinated regulation of *FT* and *TSF* expression by PIFs and CO in response to low R/FR[24] (Figs. 2, 3). These transcriptional events likely operate in the leaf vasculature where CO is known to regulate *FT* and *TSF* expression[12,34], and the PIFs are also expressed (Fig. 2e)[11]. The mechanism we uncovered here is likely to be significant in natural environments as "florigen" genes *FT* and *TSF*, as well as *PIF4*, were identified as genes underlying regulation of flowering time and shade-avoidance response in nature[42,63–65]. Our study and further deciphering the mechanism(s) regulating neighbor proximity-induced reproduction may also be relevant to increase yields on increasingly restricted agricultural land.

## Methods

**Plant material**. All experiments were performed using *A. thaliana* Columbia (Col-0) ecotype. The mutants *phyB-9*[66,67], *pif4-101*[4], *pif5-3*[68], *pif7-1*[69], *pif7-2*[69], *co-101*[34], *ft-10*[70], *tsf-1*[12], and the *pPIF4::PIF4-3HA* line[39] have been previously characterized. All experiments were performed using *pif7-1* allele, except those performed using *pif7-2/pPIF7::PIF7-3HA-tPIF7*. Mutant lines were confirmed by genotyping using oligonucleotides listed in Supplementary Table 1.

**Seeds preparation and growth conditions**. Seeds were surface sterilized with 70% ethanol and stratified for 3 days at 4 °C in darkness. For all flowering experiments seeds were germinated and seedlings de-etiolated for 5 days in high R/FR ratio (R/FR ~1.20) and on the sixth day low R/FR ratio treatment was started using supplemental FR light (R/FR ~0.20) or plants kept in high R/FR ratio (Supplementary Fig. 1). Low R/FR ratio is achieved using supplemental far-red light from light-emitting diodes light sources. Experiments were performed in plant growth incubator model AR-41L (CLF Plant Climatics) at ~22 °C with 70–80% humidity and 1:1 of Cool White and Gro-Lux Wide Spectrum fluorescent light (Phillips), with photosynthetically active radiation (PAR) of 200–220 μmol m$^{-2}$ s$^{-1}$ (Supplementary Fig. 1). Temperature was monitored using Thermochron iButtons (Maxim Integrated Products) placed at the rosette level and light spectrum and fluence was generated using Ocean Optics USB2000+ spectrometer (Supplementary Fig. 1). LD and SD photoperiod are defined as 16 h light/8 h dark and 9 h light/15 h dark, respectively.

**Flowering time phenotyping and statistical analysis**. Flowering time was scored either as total leaf number (rosette and cauline leaves) after bolting or days to flowering after sowing. Statistical analysis of flowering phenotyping was performed using analysis of variance (ANOVA) with post hoc Tukey's honestly significant difference (HSD) test using R software package and statistically significant differences ($p < 0.01$) are represented as different letters. Representative plant images were edited using Adobe Photoshop.

**Constructs cloning**. The oligonucleotides and constructs described are listed in the Supplementary Table 1. PCR amplifications were performed using Phusion High-Fidelity DNA Polymerase (New England Biolabs), unless indicated otherwise. *Arabidopsis* transformation was performed using *Agrobacterium tumefaciens* GV3301 strain by floral dip[71].

*PIF4* (−1 to −2102), *PIF5* (−1 to −2052), and *PIF7* (−1 to −3234) promoter fragments were amplified by PCR using KAPA Taq DNA polymerase (Kapa Biosystems) from MIFL8 and F17J16 BACs, and Col-0 genomic DNA, using the oligos SL-128/SL-129, SL-132/SL-133, and MK-207/MK-208, respectively. PCR-amplified fragments were restriction digested and cloned into pCB-308 *Bam*HI sites to create the vectors pAM-01 (*pPIF4::GUS*) and pAM-03 (*pPIF5::GUS*), and *Bam*HI and *Xba*I to create the vector pMK-09 (*pPIF7::GUS*). Constructs were transformed in *Arabidopsis thaliana* Col-0 ecotype.

Genomic *PIF7* fragment corresponds to −3185 to +1958 bases relative to the start codon. For cloning *pPIF7::PIF7-3HA:tPIF7*, genomic fragments were PCR amplified using Phusion polymerase (Thermo Fisher) from genomic DNA using the oligonucleotides oVCG-165/oASF-25 and oVCG-168/oASF-26, and 3HA tag amplified from pCF-402[4] with the oligonucleotides oVCG-166/oVCG-167. PCR fragments were purified and used for In-Fusion Cloning (Takara) into pKGW vector[72] previously digested with *Eco*RV to remove gateway cassette and further linearized with *Xba*I to generate the construct pASF-02.

For cloning maltose-binding protein (*MBP*) expression construct, the sequence corresponding to PIF7 bHLH domain (Supplementary Table 2) was amplified with the oligonucleotides oVCG-226/oVCG-192 from the complementary DNA (cDNA) clone TOPO-U09-H06 and cloned into pMAL-c2 using *Sac*I/*Xba*I restriction sites to generate the construct pVG-24.

For cloning the reporter construct *pTSF::fLUC* the *TSF* promoter region (−1 to −1551 bp) was amplified from the BAC clone F9F13 using the oligonucleotides oVCG-440/oVCG-441 and cloned into pGREENII-0800-LUC[73] *Xho*I/*Not*I restriction sites to generate pVG-55. For *pTSF 1xmut::fLUC* and *pTSF 3xmut::fLUC* a mutant fragment of the *TSF* promoter (−206 to −1060) was synthesized (Eurofins, Supplementary Table 2) to create the vector pEX-128_pTSF 3xmut. The *pTSF 1xmut::fLUC* (pVG-84) was generated by restriction digestion of pEX-128_pTSF 3xmut with *Pac*I/*Spe*I and cloning the purified fragment into pVG-55 (*pTSF::fLUC*). Full-length *pTSF 3xmut::fLUC* was created by PCR to amplify partial fragments using the oligos oVCG-374/oVCG-559 and oVCG-560/oVCG-381 from BAC clone F9F13 (pTSF WT), and oVCG-378/oASF-73 from pEX-128_pTSF 3xmut. Finally, the complete *pTSF 3xmut* (−1 to −1551) was PCR amplified with oVCG-440/oVCG-441 and cloned into pGREENII-0800-LUC *Xho*I/*Not*I sites.

**RNA isolation and quantitative RT-PCR**. Total RNA was extracted using QIAGEN Plant RNeasy kit and cDNA synthesis was carried out using 1 μg RNA after an on-column DNAseI digestion (RNAse-Free DNase Set, Qiagen). Reverse transcription (RT) was performed using Superscript II Reverse Transcriptase (Invitrogen, Life Technologies) with random oligonucleotides. Quantitative real-time PCR (qPCR) was performed in three biological replicates with three technical replicates for each sample using the QuantStudio 6 Flex Real-Time PCR System

(Applied Biosystems). Expression data was normalized against *UBC* and *YSL8* reference genes and relative expression analyses estimated Biogazelle qBase software. Gene-specific oligonucleotides used for qPCR reactions and respective efficiencies are listed in Supplementary Table 1.

**GUS staining**. For GUS staining reactions, samples were fixed with ice-cold 90% acetone for 30 min followed by washing twice with 50 mM NaPO$_4$ buffer (pH 7.2). Samples were incubated with staining solution (50 mM NaPO$_4$, 0.5 mM potassium ferricyanide, 0.5 mM potassium ferrocyanide, 0.1% Triton X-100 and 2 mM X-gluc) overnight at 37 °C in dark. De-staining was performed with ethanol series at room temperature until samples cleared.

**Dual-luciferase transient expression in *Nicotiana benthamiana***. *Agrobacterium tumefaciens* strain GV3301 was transformed separately with effector and reporter constructs and plated on solid YEP medium supplemented with appropriate antibiotics. After inoculating 30 mL of YEP medium with 1 mL of overnight pre-cultures, cells were incubated at 30 °C with vigorous agitation for 16 h. *Nicotiana benthamiana* leaves were infiltrated with a mixture of effector (final optical density at wavelength (OD$_{600}$) = 0.3) and/or reporter (final OD$_{600}$ = 0.1) cells along with P19 silencing suppressor (final OD$_{600}$ = 0.05) using a needleless syringe and infiltrated plants incubated in plant growth room (LD, high R/FR ratio, at 22–23 °C). Three days after incubation, two leaf discs were harvested in liquid nitrogen at zeitgeber (ZT) 8–10 for each biological replicate. After tissue lysis, dual-luciferase assay was performed using Dual-Luciferase Reporter Assay System (Promega). In short, after adding 300 μL of 1× Passive Lysis Buffer, samples were briefly vortexed for 10–15 s and spinned for 15 min at 14,000 × g at 4 °C. Ten microliters of supernatant was mixed with 40 μL luciferase substrate and chemiluminescence was measured using GloMax 96 Microplate Luminometer (Promega). The second chemiluminescence measurement was performed after adding 25 μL of 1× Stop&Glo reagent. Relative luciferase activity corresponds to the average firefly/renilla ratio from four independent infiltrations.

**Mapping PIF4 ChIP-seq reads**. The original raw data from ref. [10] obtained from the short read archive:

    GSM1665427 ChIP-seq-PIF4_FLAG_Rep1
    GSM1665428 ChIP-seq-PIF4_FLAG_Rep2
    GSM1665429 ChIP-seq-PIF4_FLAG_Rep3
    GSM1665430 ChIP-seq-PIF4_IgG_Rep1
    GSM1665431 ChIP-seq-PIF4_IgG_Rep2

Reads were first trimmed using TrimGalore (https://github.com/FelixKrueger/TrimGalore; v. 0.3.7; -q 30 --length 15) and mapped using bwa (ref. [74]; v 0.7.15) against the TAIR10 reference genome. PCR duplicates were marked using picard tools (http://broadinstitute.github.io/picard; v.2.9.0) and samtools (ref. [75]; v.1.3). The replicate 2 was chosen for the visualization of the peak areas and its mapped reads converted to binary wiggle format using bam2wig (https://github.com/MikeAxtell/bam2wig) and wigToBigWig (Bio-BigFile-1.01). Processed data, TAIR11 annotation, and published ChIP-seq peaks from ref. [10] were visualized using Integrative Genome Viewer[76] and manually edited using Adobe Illustrator.

**Western blot analysis**. For protein extraction, 15 seedlings were harvested in liquid nitrogen, ground in extraction buffer (125 mM Tris, pH 6.8, 4% sodium dodecyl sulfate (SDS), 20% glycerol, 0.02% bromophenolblue, 10% β-mercaptoethanol), and heated at 95 °C for 7 min. After centrifuging for 5 min at 15,000 × g at 4 °C, samples were separated using SDS-polyacrylamide gel electrophoresis and transferred to nitrocellulose membrane 0.45 μm (Bio-Rad). Blots were probed with anti-HA coupled with horseradish peroxidase (HRP) (1:2000; Roche, cat. 12013819001) and polyclonal anti-DET3 (1:20,000) antibody. HRP-conjugated anti-rabbit immunoglobulins (1:5000, Promega, cat. W4011) were used as secondary antibodies. Chemiluminescence signals were obtained with Immobilon Western Chemiluminescent HRP Substrate (Millipore) on an ImageQuant LAS 4000 mini (GE Healthcare). Relative intensities correspond to the average of HA/DET of three biological and at least two technical replicates obtained with the software Image Studio Lite. Uncropped Western blots are shown on Supplementary Figs. 16 and 17.

**Protein-binding microarray**. Protein-binding microarray was performed using *Escherichia coli* cultures expressing recombinant MBP_PIF7-bHLH proteins (Supplementary Table 2), synthesis of double-stranded microarray, and immunological detection of DNA–protein[77]. In summary, microarrays were scanned in DNA Microarray Scanner (Agilent Technologies) at 2 μm resolution and quantified with Feature Extraction 9.0 software (Agilent Technologies). Normalization of probe intensities and calculation of enrichment scores (*E*-scores) of all the possible 8-mers were carried out with the PBM Analysis Suite[78]. Perl scripts were modified to adapt them to different custom microarray dimensions and Feature Extraction input files[79]. Data of MBP-PIF4-bHLH and MBP-PIF5-bHLH have been previously published and are shown for comparison[8].

**Statistical analysis**. We performed one-way ANOVA (aov) and computed Tukey's HSD (HSD.test) [agricolae package] using the R software.

**Reporting summary**. Further information on research design is available in the Nature Research Reporting Summary linked to this article.

## Data availability

The authors declare that the data supporting the findings of this study are available within the manuscript and its supplementary files or are available from the corresponding author upon request. Raw data for all experiments is provided in a Source Data file.

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

## Acknowledgements

We thank Koji Goto for *pFT::GUS* line and *co-101* seeds, Andrea Maran, Séverine Lorrain, and Markus Kohnen for genomic *pPIF5::PIF5-3HA* and *pPIF4/5/7::GUS* lines, Prof. Markus Schmid for *ft-10* mutant, Prof. Hongtao Liu for *pGREENII-0800* vector, Dr. Rodrigo S. Reis for assistance with dual-luciferase assays, Genomic Technologies Facility (GTF) for assistance with RT-qPCR assays, Frédéric Schütz (SIB) for statistical advice, Dr. Adriana Arongaus, Dr. Martina Legris, and Dr. Olivier Michaud for critical reading of the manuscript. Work in the Fankhauser lab is funded by the University of Lausanne and grants from the Swiss National Science Foundation (no. 310030B_179558 and CRSII3_154438). V.C.G. was supported by EMBO long-term fellowship (ALTF 293-2013).

## Author contributions

Experimental design by V.C.G. and C.F. Experimental work and data analysis by V.C.G., A.-S.F., M.T., A.G., J.M.F.-Z, E.S.-S. and R.S. Wrote the manuscript V.C.G. and C.F.

## Additional information

**Competing interests:** The authors declare no competing interests.

