## [Peer Review File · Nature Communications]

Reviewers' comments:

Reviewer #1 (Remarks to the Author):

This manuscript addresses a key unanswered question in plant photobiology. The role of PIF transcription factors in the control of low R:FR-mediated stem elongation has been extensively explored, yet little is known about how the same signal promotes flowering. Here, Galvão and colleagues describe a clear role for PIF4, PIF5 and PIF7 in this process, via regulation of FT and TSF. Supporting evidence is provided via parallel analyses of the phyB mutant. Direct binding of PIFs to PBE boxes in the TSF promoter is reported *in vitro*. Although the findings are unsurprising, they are novel and make an important contribution to the shade avoidance literature. The paper is very well written. Data are clearly presented and support conclusions. I have two main issues I would like the authors to address:

1. In Figure 3h, the authors present time course analyses of PIF7 protein levels following transfer to low R:FR. A single band is shown. In published literature, PIF7 is shown to exist in 2 forms, a higher molecular weight phosphorylated form in white light, which is converted to a lower molecular weight dephosphorylated form in low R:FR, although this study was performed at early time points (Li et al, 2012. *Genes. Dev*). The blots in Fig.3 are cropped quite tightly so the presence/absence of a higher molecular weight band at early timepoints can't be confirmed. Can the authors show uncropped blots in the supplementary data and include greater description/discussion of PIF7 protein regulation in the results and discussion?

2. The paper provides a model of flowering for PIFs through FT and TSF, yet only PIF-TSF promoter interaction is investigated. Can the authors at least expand figure 4b to show PIF4 ChIP-seq reads mapped to the FT promoter.

Some additional minor points:

3. The *pif7* mutant is clearly also late flowering in high R:FR. Can this be highlighted on page 4.

4. Plant images should display scale bars.

5. Figure 1. The legend should highlight that plants were grown for 5 days in high R:FR before transfer to low R:FR.

6. Figure 2 legend should refer to d and e (not a and b) for GUS images.

7. Page 9- the authors refer to 2 peaks of FT transcript in response to low R:FR (Fig 3a), but there appears to be a third peak during the night.

8. Figure 3. Can the authors include the sizes of tagged PIFs on blots.

9. Figure 3. PIF5 and PIF7 should be in italics in the legend.

10. Figure 4a is a little hard to follow. Labels highlighting different motifs along the x axis and some description of E-score in the legend would help.

Reviewer #2 (Remarks to the Author):

In this manuscript, they authors showed that in response to shade signal caused by neighbor proximity, the changes in R/FR ratios (low R/FR) causes inactivation of phyB and accumulation of several PIF proteins (PIF4, PIF5 and PIF7) at the end of day in Arabidopsis, and that these proteins

could directly bind to and activate the expression of the floral induction gene TSF (and probably FT as well), and eventually leading to earlier flowering. These findings provide much new insights into the molecular mechanisms regulating flowering time by light quality, and nicely complement our advanced understanding on light quality regulation of elongation growth. The manuscript was written in a concise manner and neatly organized, with high quality data supporting their conclusions.

A few suggestions and comments for further improvement of this study are listed below:

1. For the general readers in plant science, it might be better to give a more clear introduction on the key players of this study, the PIF proteins (their basic molecular characteristics, biological roles and functional mechanisms). Also please articulate more clearly that they mean by saying that "prior identification of FT, TSF and PIF4 as loci in flowering time regulation in Nature".
2. In Figure 1d and 1e, they showed that under SD in high R/FR, the early flowering phenotype of phyB mutant was suppressed in the phyBpif457 mutant, indicating that these PIF genes act downstream of PHYB and are essential for the early flowering phenotype of phyB mutant under non-inductive photoperiod conditions, It will be best to present data using plants grown under low R/FR conditions to substantiate their claim that "PIF4, PIF5 and PIF7 act genetically downstream of phyB to control low R/FR-induced flowering".
3. They showed that the *ft* and *tsf* single mutants responded strongly to low R/FR (supplementary fig 5a-5d), and the *phyb tsf* double mutant flowered as the *phyB* mutant (supplementary fig 5e). Thus, they inferred that "neither FT nor TSF were required for early flowering". I had some difficulty understanding the precise meaning of this statement. Could the authors articulate their logics more clearly? Do the data argue a sufficient role of TSF alone to mediate the early flowering phenotype of the *phyB* mutant or do FT and TSF have partially redundant role in promoting flowering in response to low R/FR? It will be nice to have data of *phyb ft tsf* triple mutant in this analysis.
4. Through a set of higher order mutant analysis, they showed that PIFs also regulate flowering time through a CO-FT/TSF-independent pathway. This is nice and opens a new avenue for future research. It would be appreciated if the authors could provide some speculations on this point in the discussion section.
5. Fig 2 e and 2f seem not match with the Figure legend (did I miss anything here?) How old are the adult plants?
6. What "diel expression" or "diel time course" mean? Does it mean "diurnal"?
7. In Figure 3b-e, the different colored lines should be labeled clearly in each panel.
8. Does the promoter of FT has predicted G-ox or PBE box? Please explain why decided to focus on TSF, not FT (the more predominant player)? Also, It will be nice to have other in vitro or in vivo data supporting the direct binding of PIF7 to TSF promoter using yeast one-hybrid, EMSA assay or ChIP-PCR assays. Also, can the authors discuss why FT and TSF exhibit distinct induction patterns by low R/FR under diurnal conditions if both of them are direct targets of PIF proteins or alternatively discuss what could be the physiological relevance?
9. Please discuss the new findings of PIF regulating flowering in the context of known known regulatory network of FT (and TSF) by other environmental cues or transcriptional factors (such as CO).

Reviewer #3 (Remarks to the Author):

In this article Galvão et al found that PIF7, redundantly with PIF4 and PIF5, plays a role in low R/FR induced early flowering through regulation of FT and TSF gene expression. The authors claimed that the amounts of these PIF proteins are increased in the afternoon of low R/FR compared to the high R / FR condition, which led to an increase in the levels of FT and TSF mRNA. To prove this, the authors mapped PIF4 binding sites on the TSF locus using ChIP-seq and performed transient dual-luciferase assays in tobacco to confirm the ChIP-seq result.

The overall impression is that the data presented in this manuscript is not enough to support their claim.

Major points

1. The authors showed that PIF7 is the major regulator in low R/FR induced flowering among PIF proteins being used in this manuscript. However, they did not elucidate the molecular biochemical characteristics of PIF7 involved in the induction of FT and TSF expression. Instead, it appears that the authors have focused on the functions of PIF4 and 5, although the flowering phenotype of *pif45* mutant is marginal in low R/FR. They also showed PIF4, not PIF7, ChIP-seq on the TSF locus, not the FT locus, although TSF mRNA levels are not changed in the *pif45* mutant.

2. The correlation between stability changes in PIFs and FT/TSF expression is not clear. I don't understand the way that displays the relative amount of PIF proteins between two conditions. Even though the absolute amount of the proteins is low, the relative protein abundance peaks. How relevant the low amount of PIF proteins to FT and TSF induction?

3. The authors keep claiming that PIFs regulates FT and TSF expression in the vasculature, especially phloem, but I don't see any supportive data at all.

4. There are lots of flaws throughout the manuscript including many mistakes in a figure and figure legends.

For incidence, protein should be non-italic (figure 3 f to h).

Protein blots don't support the graphs.

Figure 2 e and f are unnecessary. *pPIF7*:GUS data is not provided.

And so on...

Response to reviewers' comments NCOMMS-18-33384

We would like to thank the reviewers for their advice and constructive comments. Our answers are **in bold**.

Before answering to the specific questions here is a summary of the new figures included in the revised manuscript:

- Figure 2e, *PIF7* expression in the leaf vasculature
- Supplementary Figure 5, flowering of *phyBpif* mutants in low R/FR.
- Supplementary Figure 6a, flowering of the *phyB ft tsf* triple mutant.
- Supplementary Figure 14, short-term regulation of PIF7 protein by low R/FR
- Supplementary Figure 15, plotted PIF4 ChIP-seq. data on the *FT* locus

We have also modified figures for clarification purposes.

- Figure 3 for which we now show the blots on Supplementary Figure S13
- Figure 4a

Reviewer #1 (Remarks to the Author):

1. In Figure 3h, the authors present time course analyses of PIF7 protein levels following transfer to low R:FR. A single band is shown. In published literature, PIF7 is shown to exist in 2 forms, a higher molecular weight phosphorylated form in white light, which is converted to a lower molecular weight dephosphorylated form in low R:FR, although this study was performed at early time points (Li et al, 2012. *Genes. Dev*). The blots in Fig.3 are cropped quite tightly so the presence/absence of a higher molecular weight band at early time points can't be confirmed. Can the authors show uncropped blots in the supplementary data and include greater description/discussion of PIF7 protein regulation in the results and discussion?

Answer: We have included a blot (new Supplementary Figure S14) showing that as previously observed in (Li et al., 2012 – *Genes & Development*) PIF7 migrates as two isoforms and that in low R/FR there is more of the slower (presumably phosphorylated) isoform. This is now also discussed in the revised manuscript.

Figure 3 was already very busy, we therefore removed the western blots from this figure and show them as Supplementary figure S13. Uncropped versions of Supplementary Figure S13 and S14 are shown on Supplementary Figure S16 and S17.

2. The paper provides a model of flowering for PIFs through FT and TSF, yet only PIF-TSF promoter interaction is investigated. Can the authors at least expand figure 4b to show PIF4 ChIP-seq reads mapped to the *FT* promoter.

Answer: Examination of the ChIP-seq dataset from Pedmale et al., 2016 - *Cell*, shows that there is a major PIF4 binding site containing G- and PBE-boxes 1.5 kb 3' of the

***FT* stop codon. Interestingly, this region is highly conserved amongst Brassicaceae (Zicola et al., 2019 – Nature Plants). We now present a new figure with this data (Supplementary Figure S15) and briefly discuss it. See also our response to point 8 of reviewer 2.**

Some additional minor points:

3. The *pif7* mutant is clearly also late flowering in high R:FR. Can this be highlighted on page 4.

Answer: This observation is now included in the results section and discussed in the revised manuscript.

4. Plant images should display scale bars.

Answer: Scale bars have now been included in the figures.

5. Figure 1. The legend should highlight that plants were grown for 5 days in high R:FR before transfer to low R:FR.

Answer: This important clarification has now been added to the figure legends.

6. Figure 2 legend should refer to d and e (not a and b) for GUS images.

Answer: In the modified figure 2, the promoter GUS lines (*FT*, *PIF4*, *PIF5* and *PIF7*) are now all grouped as panel e.

7. Page 9 - the authors refer to 2 peaks of *FT* transcript in response to low R:FR (Fig 3a), but there appears to be a third peak during the night.

Answer: We have now indicated in the text that indeed there is a third smaller peak of *FT* and *TSF* expression.

8. Figure 3. Can the authors include the sizes of tagged PIFs on blots.

Answer: Western blots including the position of MWM are now included in new supplementary Figures 13, 14, 16 and 17.

9. Figure 3. *PIF5* and *PIF7* should be in italics in the legend.

Answer: The legend was corrected in the revised manuscript

10. Figure 4a is a little hard to follow. Labels highlighting different motifs along the x axis and some description of E-score in the legend would help.

Answer: We have modified Figure 4a and included an extended explanation of the E-

score in the legend.

Reviewer #2 (Remarks to the Author):

1. For the general readers in plant science, it might be better to give a more clear introduction on the key players of this study, the PIF proteins (their basic molecular characteristics, biological roles and functional mechanisms). Also please articulate more clearly that they mean by saying that “prior identification of FT, TSF and PIF4 as loci in flowering time regulation in Nature”.

Answer: We have now included a short introduction about the PIFs focusing on their role in the response to shade cues. Regarding the second part of the comment we tried to clarify the meaning of the last phrase of the abstract by changing “nature” with “natural conditions” as the studies we refer to (and discuss later in the paper) are about natural variation amongst Arabidopsis accessions and how changes in these genes modulate flowering time.

2. In Figure 1d and 1e, they showed that under SD in high R/FR, the early flowering phenotype of *phyB* mutant was suppressed in the *phyBpif457* mutant, indicating that these PIF genes act downstream of PHYB and are essential for the early flowering phenotype of *phyB* mutant under non-inductive photoperiod conditions, It will be best to present data using plants grown under low R/FR conditions to substantiate their claim that “PIF4, PIF5 and PIF7 act genetically downstream of phyB to control low R/FR-induced flowering”.

Answer: We have now included new experiments to respond to this insightful comment as indeed it was previously shown that although phyB is an important regulator of flowering in response to shade other phytochromes also contribute (e.g. Halliday et al., 1994 – Plant Physiology). This new data presented in Supplementary Figure S5 shows that PIF4/5/7 indeed act downstream of phyB in low R/FR to control flowering time.

3. They showed that the *ft* and *tsf* single mutants responded strongly to low R/FR (supplementary fig 5a-5d), and the *phyB tsf* double mutant flowered as the *phyB* mutant (supplementary fig 5e). Thus, they inferred that “neither FT nor TSF were required for early flowering”. I had some difficulty understanding the precise meaning of this statement. Could the authors articulate their logics more clearly? Do the data argue a sufficient role of TSF alone to mediate the early flowering phenotype of the *phyB* mutant or do FT and TSF have partially redundant role in promoting flowering in response to low R/FR? It will be nice to have data of *phyB ft tsf* triple mutant in this analysis.

Answer: To clarify this part of our argument we followed the reviewers advise and now present new data that includes the flowering time phenotype of *phyB ft tsf* triple mutants in high and low F/FR (new panel E of Supplementary Figure S6). We believe that with the associated text changes this clarifies the manuscript. Similarities and difference between early flowering triggered by low R/FR vs. in a *phyB* mutant are also mentioned in the discussion.

4. Through a set of higher order mutant analysis, they showed that PIFs also regulate flowering time through a CO-FT/TSF-independent pathway. This is nice and opens a new avenue for future research. It would be appreciated if the authors could provide some speculations on this point in the discussion section.

Answer: As suggested by the reviewer we have now expanded the discussion regarding the mechanisms through which the PIFs could regulate flowering in response to shade cues.

5. Fig 2 e and 2f seem not match with the Figure legend (did I miss anything here?) How old are the adult plants?

Answer: In our revised figure 2, all promoter GUS lines are now shown in panel e (including new data for PIF7).

6. What “diel expression” or “diel time course” mean? Does it mean “diurnal”?

Answer: In the literature authors sometimes use the term diurnal when referring to a pattern observed when organisms are grown in day:night conditions. The term diurnal is incorrect in this context as diurnal refers to the behavior of the organism (diurnal or nocturnal). The appropriate term is diel.

7. In Figure 3b-e, the different colored lines should be labeled clearly in each panel.

Answer: We have modified figure 3 to better highlight what the different bar graphs and lines correspond to. Moreover, we moved the western blots to Supplementary Figure S13. See also our answers to comments 1 and 8 of reviewer 1.

8. Does the promoter of FT has predicted G-ox or PBE box? Please explain why decided to focus on TSF, not FT (the more predominant player)? Also, It will be nice to have other in vitro or in vivo data supporting the direct binding of PIF7 to TSF promoter using yeast one-hybrid, EMSA assay or ChIP-PCR assays. Also, can the authors discuss why FT and TSF exhibit distinct induction patterns by low R/FR under diurnal conditions if both of them are direct targets of PIF proteins or alternatively discuss what could be the physiological relevance?

Answer: As also requested by reviewer 1 we now present and discuss the ChIP-seq data for *FT* from Pedmale et al., 2016 – Cell in Supplementary Figure S15. Based on this data it appears that the regulation of *TSF* is potentially simpler than the one of *FT* (see also Zicola et al., 2019– Nature Plants that is now cited). PIF regulation of *FT* expression has been studied and appears to be a somewhat controversial issue. Kumar et al., 2012 propose a PIF4 binding peak in the *FT* promoter that was not observed by others. Moreover the importance of PIF4 for temperature regulation of expression initially presented in Kumar et al., 2012 was not observed in subsequent studies (Galvão et al., 2015 in Plant Journal vol 84, pages 949-62; Seaton et al., 2015 in Mol

Systems Biology 11, 776; Fernandez et al., 2016 in Plant Journal vol 86, pages 426-60; Sureshkumar et al., 2016 in Nature Plants vol 2 article number 16055). In contrast, much less is known about the regulation and function of *TSF* and our data clearly shows that *TSF* (with *FT*) is important to regulate flowering time in response to a reduction of the R/FR ratio (e.g. Supplementary Figure S6e). We have clarified this in the revised manuscript.

The *TSF* promoter and in particular the PIF4 binding peak identified by Pedmale et al., 2016 contains no G-box but a PBE box (Figure 4). We have performed Y1H and EMSA assays with parts of the *TSF* promoter and PIF7, which has the strongest effect of low R/FR-regulated flowering. This data provided no evidence of PIF7 binding to the *TSF* promoter, which is actually consistent with data presented in Figure 4a showing that the affinity of PIF7 to a PBE element is weak. As positive controls for these experiments we used the *YUC8* promoter, which contains 2 G-boxes that are bound with higher affinity by PIF7 (Figure 4a). We note that PBE elements are also present in a many PIF5 ChIP peaks although as for PIF7, PIF5 binds very weakly to a PBE-box *in vitro* (Figure 4a and Hornitschek et al., 2012). Nevertheless, our data in *Nicotiana* shows that in plants PIF7 and PIF4 regulate *TSF* expression and this regulation requires the PBE element (see figure 4c and d). One possible interpretation of our data is that binding of PIF7 to the *TSF* promoter requires additional factors that are not present in yeast or in an EMSA assay. CO could be such a factor and it will be interesting to test whether CO and PIFs cooperate to bind to DNA. Given the enormous difficulty to show binding of CO to promoters *in vivo* this is an interesting and also challenging avenue. We have not included the negative results discussed above in the revised manuscript because it is always difficult to fully interpret negative data. However, if the reviewer would like to see it we would be happy to provide it.

Regarding the slower induction kinetics of *TSF* than *FT* by low R/FR (Figure 2a,b), we can only speculate. One possibility would be that this allows to trigger the flowering response more efficiently in response to a prolonged shade treatment and not by a short shade episode. However, this is highly speculative and we think that further experimentation would be required before discussing this point in the paper.

9. Please discuss the new findings of PIF regulating flowering in the context of known regulatory network of FT (and TSF) by other environmental cues or transcriptional factors (such as CO).

Answer: This point is potentially related to the previous point. We now propose more explicitly that the combined action of PIFs and CO allows low R/FR-induced acceleration of flowering (4th paragraph of the discussion). This hypothesis is fully consistent with the fact that low R/FR accelerates flowering in long but not short days.

Reviewer #3 (Remarks to the Author):

1. The authors showed that PIF7 is the major regulator in low R/FR induced flowering among PIF proteins being used in this manuscript. However, they did not elucidate the

molecular biochemical characteristics of PIF7 involved in the induction of FT and TSF expression. Instead, it appears that the authors have focused on the functions of PIF4 and 5, although the flowering phenotype of *pif45* mutant is marginal in low R/FR. They also showed PIF4, not PIF7, ChIP-seq on the TSF locus, not the FT locus, although TSF mRNA levels are not changed in the *pif45* mutant.

Answer. We agree with the reviewer that our genetic data is consistent with PIF7 playing a dominant role in low R/FR-induced flowering. However, our data also clearly show that PIF4 and PIF5 contribute to the response. Please have a look at Figures 1b, 1c and S2 and compare *pif7* with *pif4pif5pif7*. Our conclusion that multiple PIFs contribute to low R/FR-induced flowering is also fully consistent with the suppression of precocious flowering of *phyB* shown on figure 1e (compare *phyB pif7* with *phyBpif4pif5pif7*). Moreover, these flowering time phenotypes are matched by the role played by PIF4, PIF5 and PIF7 regarding the regulation of *FT* and *TSF* expression (Figure 2c and d and S9). Again, please compare *pif7* with *pif4pif5pif7*. The regulation of RNA and protein expression is presented for all three PIFs (Figure 3 and S13). Finally, the functional importance of both PIF4 and PIF7 for the regulation of *TSF* expression is presented on Figure 4c and d, respectively.

2. The correlation between stability changes in PIFs and *FT/TSF* expression is not clear. I don't understand the way that displays the relative amount of PIF proteins between two conditions. Even though the absolute amount of the proteins is low, the relative protein abundance peaks. How relevant the low amount of PIF proteins to FT and TSF induction?

Answer. We have modified Figure 3f-h in order to improve our presentation and clarify the meaning of the different bars and lines. While the absolute levels of PIFs are lower at the end of the day, the difference between high and low R/FR are highest at this time of the day (dashed red bar on the figures). This corresponds to the time when we observe a big difference in *FT* and *TSF* expression between high and low R/FR that depends on the PIFs (Figure 3 a, b).

3. The authors keep claiming that PIFs regulates FT and TSF expression in the vasculature, especially phloem, but I don't see any supportive data at all.

Answer. We now provide a revised version of figure 2. On panel e we show that PIF7, PIF4 and PIF5 are all expressed in the vasculature. This data is fully consistent with tissue specific expression data that also demonstrated expression of *PIF7*, *PIF4* and *PIF5* in the vasculature (Kim et al., 2018 – Plant Physiology and You et al., 2019 – Plant Cell).

4. There are lots of flaws throughout the manuscript including many mistakes in a figure and figure legends.

Answer. We have corrected a number of typos and other mistakes

For incidence, protein should be non-italic (figure 3 f to h).

Answer. This has now been corrected

Protein blots don't support the graphs.

Answer. The graphs represent the quantification of the blots. We suspect that the reviewer did not understand the meaning of the red dotted line. We hope that the revised version of the figure now clarifies this.

Figure 2 e and f are unnecessary. pPIF7:GUS data is not provided.

Answer. We have modified Figure 2 and now include data for *pPIF7:GUS*.

REVIEWERS' COMMENTS:

Reviewer #1 (Remarks to the Author):

The authors have addressed my concerns and significantly improved the manuscript. A few very minor areas for improvement are as follows:

1. Page 3, line 65, 'We start to understand...' should read 'We have started to understand..'
2. Page 13, line 241, Figure 3a,b should also be referenced for the statement starting 'Multiple levels of PIFs...'
3. Scale bars have been added to Supplementary Figures 8 and 11, but the length of this is not stated in the legend.

Reviewer #2 (Remarks to the Author):

The authors have satisfactorily addressed my previous concerns.

Reviewer #3 (Remarks to the Author):

The revised manuscript improved some parts but still not satisfy the main criticism. As previously suggested, showing the relationship between PIFs and *FT* expression in low R/FR is crucial in this paper, not only by just *TSF*. Although PIF4 binds to the *FT* locus, who knows it happens all the time. Not only this reviewer but also others might would like to see if PIFs directly regulate *FT* transcription in your conditions, not in someone's.

As shown in Fig 3 a and b, *FT* and *TSF* expression shows morning, evening and night peaks. Substantial differences in the accumulation of PIF proteins between high and low R/FR conditions are observed in the evening. However, *FT* and *TSF* expression is consistently low in the *pif457* mutant throughout the day. At least this kind of high impact journal, a molecular mechanism(s) should be addressed.

In addition, the authors might misunderstand the previous comment that displaying the red dot lines in Fig 3 f, g, h is not ideal. Even changes in protein amounts between two conditions are observed, it is hard to suspect biological relevance if the absolute protein amount is relatively low, unless proving it. Also, comparing protein amounts from two different blot membranes is not recommended shown in Supple Fig 13, because we are not sure that we always get the same signal intensity.

Response to reviewers' comments NCOMMS-18-33384A

We would like to thank the reviewers for their advice and constructive comments. Our answers are **in bold**.

Reviewer #1 (Remarks to the Author):

The authors have addressed my concerns and significantly improved the manuscript. A few very minor areas for improvement are as follows:

We thank the reviewer

1. Page 3, line 65, 'We start to understand...' should read 'We have started to understand..'

Text corrected accordingly.

2. Page 13, line 241, Figure 3a,b should also be referenced for the statement starting 'Multiple levels of PIFs...'

Text corrected accordingly.

3. Scale bars have been added to Supplementary Figures 8 and 11, but the length of this is not stated in the legend.

Figure legend corrected accordingly.

Reviewer #2 (Remarks to the Author):

The authors have satisfactorily addressed my previous concerns.

We thank the reviewer

Reviewer #3 (Remarks to the Author):

We thank the reviewer for his suggestions

The revised manuscript improved some parts but still not satisfy the main criticism. As previously suggested, showing the relationship between PIFs and *FT* expression in low R/FR is crucial in this paper, not only by just *TSF*. Although PIF4 binds to the *FT* locus, who knows it happens all the time. Not only this reviewer but also others might would like to see if PIFs directly regulate *FT* transcription in your conditions, not in someone's.

The regulation of *FT* expression and the role played by PIF transcription factors is a complex and based on previous studies about temperature-induced flowering a somewhat contentious issue (see the discussion). Future research will address the precise mechanism for this regulation.

As shown in Fig 3 a and b, *FT* and *TSF* expression shows morning, evening and night peaks. Substantial differences in the accumulation of PIF proteins between high and low R/FR conditions are observed in the evening. However, *FT* and *TSF* expression is consistently low in the *pif457* mutant throughout the day. At least this kind of high impact journal, a molecular mechanism(s) should be addressed.

We note that flowering time in the *pif* mutants is much more affected in low R/FR than in standard light conditions (e.g. Figure 1), which indicates that the PIFs are particularly important for accelerated flowering in response to neighbor proximity (low R/FR). Moreover, this conclusion is consistent with a recent report from the Imaizumi lab published in Nature Plants (Song et al., 2018). The role of PIFs in regulating *FT* and *TSF* expression reported here are obvious in low R/FR (see Figure 2a,b,c,d and Figure 3a,b that shows a clear difference in expression in low but not high R/FR) and this difference correlates with the flowering time phenotype.

In addition, the authors might misunderstand the previous comment that displaying the red dot lines in Fig 3 f, g, h is not ideal. Even changes in protein amounts between two conditions are observed, it is hard to suspect biological relevance if the absolute protein amount is relatively low, unless proving it.

The importance of timing and not only total amount of a regulatory protein is often critical. This is particularly true for processes involving circadian regulation. A model that is consistent with the former literature and data presented here is that PIFs together with CO may enhance *FT* and *TSF* expression. According to this model, high levels/activity of the PIFs late in the day when CO accumulates explains enhanced *FT* and *TSF* expression (see discussion). It will be interesting to decipher the molecular mechanisms underlying PIF-CO induced flowering.

Also, comparing protein amounts from two different blot membranes is not recommended shown in Supple Fig 13, because we are not sure that we always get the same signal intensity.

We agree with the reviewer and have not made direct comparisons of protein levels between blots but rather used normalized data from different blots to avoid this problem.